# Differences in the Shiga Toxin (Stx) 2a Phage Regulatory Switch Region Influence Stx2 Localization and Virulence of Stx-Producing *Escherichia coli* in Mice

**DOI:** 10.3390/microorganisms11081925

**Published:** 2023-07-28

**Authors:** Rama R. Atitkar, Angela R. Melton-Celsa

**Affiliations:** 1Department of Microbiology and Immunology, Uniformed Services University of the Health Sciences, Bethesda, MD 20814, USA; 2Henry M. Jackson Foundation for the Advancement of Military Medicine, Inc., Bethesda, MD 20817, USA

**Keywords:** *Escherichia coli*, Shiga toxin, phage, CI repressor, *recA*

## Abstract

Shiga toxin (Stx)-producing *Escherichia coli* (STEC) is a major cause of foodborne illness globally, and infection with serotype O157:H7 is associated with increased risk of hospitalization and death in the U.S. The Stxs are encoded on a temperate bacteriophage (*stx*-phage), and phage induction leads to Stx expression; subtype Stx2a in particular is associated with more severe disease. Our earlier studies showed significant levels of RecA-independent Stx2 production by STEC O157:H7 strain JH2010 (*stx*_2a_*stx*_2c_), even though activated RecA is the canonical trigger for *stx*-phage induction. This study aimed to further compare and contrast RecA-independent toxin production in Stx2-producing clinical isolates. Deletion of *recA* in JH2010 resulted in higher in vitro supernatant cytotoxicity compared to that from JH2016Δ*recA*, and the addition of the chelator ethylenediaminetetraacetic acid (EDTA) and various metal cations to the growth medium exacerbated the difference in cytotoxicity exhibited by the two deletion strains. Both the wild-type and Δ*recA* deletion strains exhibited differential cytotoxicity in the feces of infected, streptomycin (Str)-treated mice. Comparison of the *stx*_2a_-phage predicted protein sequences from JH2010 and JH2016 revealed low amino acid identity of key phage regulatory proteins that are involved in RecA-mediated *stx*-phage induction. Additionally, other STEC isolates containing JH2010-like and JH2016-like *stx*_2a_-phage sequences led to similar Stx2 localization, as demonstrated by JH2010Δ*recA* and JH2016Δ*recA*, respectively. Deletion of the *stx*_2a_-phage regulatory region in the wild-type strains prevented the differential localization of Stx2 into the culture supernatant, a finding that suggests that the *stx*_2a_-phage regulatory region is involved in the differential Δ*recA* phenotypes exhibited by the two strains. We hypothesize that the amino acid differences between the JH2010 and JH2016 phage repressor proteins (CIs) lead to structural differences that are responsible for differential interaction with RecA. Overall, we discovered that non-homologous *stx*_2a_-phage regulatory proteins differentially influence RecA-independent, and possibly RecA-dependent, Stx2 production. These findings emphasize the importance of studying non-homologous regulatory elements among *stx*_2_-phages and their influence on Stx2 production and virulence of STEC isolates.

## 1. Introduction

Shiga toxin (Stx)-producing *Escherichia coli* (STEC) causes an estimated 265,000 cases of gastrointestinal disease in the U.S. annually [1]. The majority of these cases are sporadic infections, but large, multi-state STEC outbreaks also occur. Almost 40% of the total number of STEC cases are linked to serotype O157:H7 isolates. In addition, serotype O157:H7 STEC are more frequently associated with outbreaks and severe disease progression in the U.S. than other STEC serotypes [2,3]. Mild STEC cases present as diarrhea and abdominal cramps, with most patients experiencing self-limiting and self-resolving symptoms. However, the development of severe gastrointestinal symptoms (including bloody diarrhea and hemorrhagic colitis) is linked to the development of hemolytic uremic syndrome (HUS), a systemic sequela of the STEC infection. Progression to HUS occurs in approximately 3–7% of STEC patients, although some outbreaks have HUS rates as high as 25% [4]. HUS patients suffer kidney damage in addition to hemolytic anemia and thrombocytopenia. Severe HUS cases may result in long-term kidney damage or death.

Severe systemic symptoms of STEC infection are mediated by Stx activity. Stx binds to Gb3 on the surface of susceptible cells and is translocated in a retrograde manner to the cytoplasm, where it targets host cell ribosomes and prevents host cell protein synthesis [5]. Toxin types Stx1 and Stx2 have the same mode of action, but are antigenically distinct. Each toxin type can be further divided into subtypes of varying toxicity; subtypes Stx2a, Stx2c, and Stx2d in particular have been associated with more severe human disease [6]. The Stx-encoding genes (*stx*) are found on a temperate lambdoid bacteriophage (*stx*-phage). Typically, the *stx* phage is integrated within the bacterial chromosome (called lysogeny), but can be induced into the lytic cycle by activated RecA (RecA*). The *stx*-phage lytic cycle involves the replication and packaging of phage genomes into phage particles that are eventually released from the cell through bacterial lysis. The *stx* genes are expressed as late genes during the phage lytic cycle; thus, increased *stx*-phage induction is associated with increased Stx expression and release [7]. In fact, the DNA-damaging antibiotic ciprofloxacin (Cip) increases RecA-dependent *stx*-phage induction and Stx production, and its use has been linked to more severe disease in STEC-infected children [8].

While RecA* is primarily responsible for lambda phage induction, low levels of induction can still be detected in *recA*-negative strains, a finding that shows that there are RecA-independent mechanisms of phage induction [9,10]. Due to the lambdoid classification of *stx*-phages, one would predict that the deletion of *recA* in *stx*-phage-containing strains would also reveal RecA-independent mechanisms of phage induction. However, a study on O157:H7 STEC strains EDL933 and 86–24 reported no detectable phage particles or Stx production in vitro in *recA* mutant strains [11]. In contrast, RecA-independent *stx*-phage induction was detected in uninduced cultures from *E. coli* K-12 *stx*-phage lysogens [12,13]. We previously found that deletion of *recA* in JH2010, a *stx*_2a_*stx*_2c_ O157:H7 STEC strain, led to only a 100-fold reduction of in vitro and in vivo cytotoxicity compared to wild-type JH2010, a result that indicates high levels of *recA*-independent Stx2 production [14].

This study aimed to further compare and contrast RecA-independent toxin production in Stx2-producing O157:H7 clinical isolates. Overall, we discovered that non-homologous *stx*_2a_-phage regulatory proteins differentially influence RecA-independent Stx production, possibly through differences in CI-mediated repression of the *stx*-phage.

## 2. Materials and Methods

Bacterial Strains and Plasmids. Bacterial cultures (1–2 mL) were grown overnight in LB-YE (10 g tryptone and 5 g NaCl per liter) at 37 °C, 250 RPM for 16–18 h. Cultures were started from 3 single colonies picked from a plate. When necessary, additional chemicals were added at the start of the culture incubation. Stx localization for cytotoxicity determination was determined by taking 1 mL of the overnight culture (whole culture) and/or centrifuging an aliquot of culture (for 30 s at 13,000 rpm, at room temperature) and collecting the supernatant fraction (supernatant). The remaining cell pellet after centrifugation was resuspended in 1 mL phosphate buffered saline (cell-associated). All samples were stored at −20 °C. Whole culture and cell-associated fractions were sonicated on ice to release intracellular/membrane-associated Stx, then centrifuged (for 30 s at 13,000 rpm, at room temperature), and stored on ice briefly before use of the supernatant in the Vero cell cytotoxicity assay. All bacterial strains used in this study are listed in Table 1.

Mouse Infection Protocol. Mouse experiments were approved by the Institutional Animal Care and Use Committee of the Uniformed Services University and conducted following recommendations from the *Guide for the Care and Use of Laboratory Animals* [17]. The 14–16 g male BALB/c mice (Jackson Laboratories) were given streptomycin water (Str, 5 g/L) for 48 h prior to infection to promote colonization, as discussed previously [18]. After a 3 h fast, mice were orally gavaged with 10^10−11^ CFU of STEC per mouse (unless specified otherwise). Infected mice remained on Str water and were monitored for clinical symptoms until 14 days post-infection.

Mice were checked for weight and survival daily from the start of Str water administration through the experiment endpoint, except on the day of infection. Fecal pellets were collected from individual infected mice at days one and three post-infection, and were homogenized in PBS. The debris was allowed to settle for approximately 15 min, then diluted. An aliquot of the diluted supernatant was plated on LB with the appropriate antibiotic selection (either 50 ug/mL Str or 30 ug/mL chloramphenicol) to determine fecal colonization (CFU/g feces). As determined historically over many studies, the colonies that grow on Str of the appropriate size are STEC [14,18,19]. Occasionally, colonies of very small size are also seen on the LB +Str, but these are not STEC. The remaining supernatant was used to determine fecal cytotoxicity (CD_50_/g feces) by Vero cell cytotoxicity assay.

Bacterial Mutagenesis. Gene deletions in STEC isolates were created using the lambda Red recombinase system for mutagenesis, as previously published [14]. Briefly, STEC isolates containing plasmid pTP1215 [20] were transformed with 500 ng of purified PCR construct containing a chloramphenicol (Cm) resistance gene (*cat*) flanked by 50 bp of homology to the gene targeted for deletion. Recombinants were selected on LB+Cm and screened for the correct size using PCR, to confirm the intended gene deletion. The primers used to create the mutagenesis PCR constructs and screen for successful deletion are included in Table 2.

Vero Cell Cytotoxicity Assay. The Vero cell cytotoxicity assay was conducted as described previously to determine the cytotoxicity of Stx-containing samples [14]. Briefly, serial dilutions of Stx-containing samples were layered on 96-well plates containing Vero cell monolayers. The plates were incubated for 48 h, then fixed with 10% buffered formalin and stained with crystal violet. The absorbance at 590 nm was read for rinsed and dried plates to calculate the CD_50_ (1/(sample dilution) needed to kill 50% of untreated Vero cells).

RT-qPCR. The Quick-RNA Miniprep Kit (Zymo Research, Irvine, CA, USA) was used to isolate RNA from overnight bacterial cultures. A total of 100 ng of RNA was converted to cDNA using the Quantitect Reverse Transcription Kit (Qiagen, Hilden, Germany). qPCR reactions and corresponding controls were set up using the SYBR Green qPCR Kit (Qiagen), the designated primer pairs, and 1.5 μg of each cDNA template. The primer pairs and qPCR conditions are listed below, in Table 2. The housekeeping gene *gyrB* was used as the endogenous control, and data for each reaction were presented as fold change of expression from *gyrB* (log_2_2^−ΔCt^).

Genomic and Protein Analyses. The *stx*_2a_-phage sequences were aligned with NCBI Nucleotide Blast and visualized using EasyFig [22]. Individual protein sequences were analyzed using Geneious Prime (version 2022.2.2), and percent nucleotide and amino acid identities were determined by the Clustal Omega tool in Geneious Prime. InterProScan (version 92.0) was used to analyze protein sequences for various protein domains within the InterPro database. The Phyre2 algorithm in normal mode predicted the secondary and tertiary structure using protein sequences and reference structures within the PDB database. Predicted protein structures were visualized using the Geneious Prime PDB viewer in Geneious Prime.

Statistical analyses. GraphPad Prism (Version 9.0.2 for Windows) was used to conduct all statistical analyses.

## 3. Results

### 3.1. RecA-Independent Production of Stx2 from STEC Strains JH2010 and JH2016

To determine if our previous observation of detectable RecA-independent levels of Stx2 production from O157:H7 strain JH2010 could be extended to another STEC strain, we deleted *recA* in O157:H7 strain JH2016. Like JH2010, the JH2016 genome contains two independent *stx*_2_-phages, one which encodes *stx*_2a_ and the other *stx*_2c_. Wild-type JH2010 and JH2016 exhibited similar whole culture, cell-associated, and supernatant cytotoxicity, Figure 1A–C. Deletion of *recA* in JH2010 and JH2016 significantly reduced the whole culture cytotoxicity from wild-type levels (2–4 × 10^5^ CD_50_/mL) to comparable levels between JH2010Δ*recA* and JH2016Δ*recA* (4–5 × 10^3^ CD_50_/mL), Figure 1A. In contrast, the cell-associated cytotoxicity was significantly reduced as compared to wild-type only from JH2016Δ*recA*, Figure 1B. Similar to the whole culture samples, the level of cytotoxicity found in the supernatant fraction of the *recA* mutant strains was reduced by about 100-fold from wild-type, Figure 1C. We were surprised to find that JH2010Δ*recA* exhibited significantly higher supernatant cytotoxicity as compared to JH2016Δ*recA*, Figure 1C. The observed difference in JH2010Δ*recA* and JH2016Δ*recA* supernatant cytotoxicity is not due to a difference in growth, because while modest differences in OD_600_ were observed, there was no significant difference in CFU/mL between cultures of the two *recA* deletion strains, Appendix A. Complementation of the JH2010Δ*recA* deletion strain in *trans* (Δ*recA/recA^+^*) restored cytotoxicity to wild-type levels in the JH2010 background, Figure 1. In contrast, complementation of the JH2016Δ*recA* strain led to significantly higher whole culture and supernatant cytotoxicity compared to wild-type JH2016, Figure 1A,C, possibly due to strain-specific differences in complementation plasmid copy number.

### 3.2. Cytotoxicity from Feces of Mice Infected with JH2010ΔrecA or JH2016ΔrecA

To compare the levels of toxin produced in vivo from the *recA* mutant strains, we infected streptomycin (Str)-treated mice orally with 10^10−11^ CFU of either strain. As expected, the *recA* mutant strains were avirulent in the mice, a result we had observed previously with JH2010Δ*recA* [14]. However, we detected a modest transient decrease in the weights of JH2010Δ*recA*-infected mice during two independent experiments, while JH2016Δ*recA*-infected mice did not experience a weight decrease, Appendix A. Unlike JH2016Δ*recA*-infected mice, we found that JH2010Δ*recA*-infected mice exhibited detectable fecal cytotoxicity on days 1 and 3 post-infection, Figure 2A. A slight, but significant, difference in fecal colonization of the two strains was observed on day 1 post-infection, but not on day 3, Figure 2B. It is unlikely that the roughly 2.5-fold difference in colonization observed on day 1 is responsible for the failure to detect toxin from the JH2016Δ*recA*-infected mice, since a similar difference in cytotoxicity was observed on day 3 when there was no significant difference in colonization between the two strains. Taken together, our results indicate that in the absence of RecA, JH2010 releases more Stx2 in vitro and produces more Stx2 in vivo as compared to JH2016.

### 3.3. Test of Potential Inducers of RecA-Independent Cytotoxicity

We next explored the effect of previously identified RecA-independent inducers of toxin expression, to determine if they altered toxin expression from JH2010Δ*recA* or JH2016Δ*recA*. The addition of the RecA-dependent inducer Cip to cultures of either *recA* deletion strain did not influence the resulting cytotoxicity, as observed previously for JH2010Δ*recA* [14]. The chelator EDTA leads to RecA-independent in vitro *stx*-phage induction and Stx production from *E. coli* K-12 *stx*-phage lysogens when included in the growth medium [12]. We found that when incubated with 50 μM EDTA, JH2010Δ*recA* cultures exhibited a modest but significant increase in cytotoxicity, as compared to the corresponding control culture, while JH2016Δ*recA* cultures did not, Figure 3. The addition of metal salts has been shown to eliminate the RecA-independent influence of EDTA on *stx*-phage induction [12], and has also been shown to influence RecA-independent induction of phage λ*^imm434^* [23], so we next added various divalent cationic salts to cultures of JH2010Δ*recA* or JH2016Δ*recA*. Compared to control cultures, the addition of each of the three salts reduced the whole culture cytotoxicity of JH2016Δ*recA* cultures as compared to the control, while only the addition of CaCl_2_ decreased the cytotoxicity of JH2010Δ*recA* cultures, Figure 3 (see # for significant difference, as compared to the control culture). We were surprised that the cytotoxicity from JH2016Δ*recA* cultures was affected by the addition of divalent cations, since we observed that the cytotoxicity from JH2016Δ*recA* was not altered by the addition of EDTA, Figure 3. Overall, we found that the addition of EDTA or metal ions differentially influenced the RecA-independent cytotoxicity of JH2010 and JH2016.

### 3.4. Infection of Mice with an Intermediate Dose of JH2010 or JH2016

The in vivo data, Figure 2, suggests that JH2010 and JH2016 have strain-specific differences in RecA-independent induction of the *stx*_2_-phage. Our previous study showed that at a dose of 10^10−11^ CFU/mouse, JH2010 and JH2016 had similarly high virulence and fecal cytotoxicity [14]. Therefore, we infected Str-treated mice with an intermediate dose of 10^6−7^ CFU/mouse to determine the influence of a lower infectious dose on the RecA-dependent colonization capacity and virulence of the strains. At the intermediate inoculum, JH2010-infected mice had significantly lower percent survival and higher day-1 fecal cytotoxicity compared to JH2016-infected mice, Figure 4A,B. The reduced virulence and toxin production was not due to differences in colonization, as both JH2010- and JH2016-infected mice had similar colonization on days 1 and 3 post-infection, Figure 4C. Combined, the mouse experiments confirmed that JH2010 has an increased capacity for RecA-independent and RecA-dependent toxin production under in vivo conditions, as compared to JH2016.

### 3.5. Comparison of the stx_2a_-Phage Sequences of JH2010 and JH2016

Taken together, our findings indicate that JH2010 and JH2016 respond to the deletion of *recA* with differential toxin localization and altered toxin production in response to environmental influences. Since we found that Stx2a rather than Stx2c is responsible for the virulence of JH2010 [14], we next compared the nucleotide homology and synteny of the *stx*_2a_-phage sequences from strains JH2010 and JH2016 using NCBI Blast, Figure 5. The pairwise alignment showed regions of either high (>90%) or low (<70%) identity between the two *stx*_2a_-phage sequences. An insertion element found downstream of the JH2016 *stx*_2a_-phage *S* and *R* genes was duplicated in JH2010, with one copy located before the *S* and *R* genes and the other copy inverted and downstream of the *S* and *R* genes. The regions of low homology between the two sequences include a ~4400 bp sequence between the recombination and replication regions and a ~1700 bp sequence between the replication and Stx regions. In particular, the *N*, *cI*, and *cro* genes and their encoded proteins shared low nucleotide and amino acid identity between JH2010 and JH2016, Appendix A. We therefore hypothesized that the *stx*_2a_-phage regulatory regions of JH2010 and JH2016 may differentially influence downstream Stx2 production.

Next, we wanted to see if the difference in RecA-independent cytotoxicity associated with JH2010 and JH2016 could be replicated in other O157:H7 Stx2-producing strains. We identified STEC isolates with *stx*_2a_-phage regulatory sequences identical to either the JH2010 or JH2016 *stx*_2a_-phage from within the PulseNet STEC Genome Reference Library or from other NCBI BioProjects. The identified isolates encoded *stx*_2a_-phages sharing >99% nucleotide identity with the *stx*_2a_-phage from either JH2010 (strains 08-3914, PA28) or JH2016 (strains 2010c-3142, TW14359), Appendix A. We created Δ*recA* mutant strains in Str^r^ isolates from the NCBI strains, and measured the in vitro cytotoxicity of the mutant strains after overnight growth in LB-YE. We found that the JH2010-like Δ*recA* mutant strains exhibited significantly higher cell-associated and supernatant cytotoxicity compared to the JH2016-like *recA* mutant strains, Figure 6A. These results suggest that differences in the *stx*_2a_-phages from JH2010 and JH2016 may be responsible for the observed phenotypes of differential toxin localization and RecA-independent cytotoxicity.

To pursue our hypothesis that differences in the *stx*_2a_-phage regulatory region contributes to the differential cytotoxicity between JH2010 and JH2016, we deleted the *stx*_2a_-phage region from *N* to *cro* in JH2010 and JH2016 (Δ*N*Δ*cI*Δ*cro*), Appendix A. We reasoned that if the Δ*N*Δ*cI*Δ*cro* deletion in both strains did not lead to differential supernatant cytotoxicity as observed between JH2010Δ*recA* and JH2016Δ*recA*, then the *N* to *cro* region is required for differences in the RecA-independent localization of toxin into the supernatant. We obtained Δ*N*Δ*cI*Δ*cro* derivative strains in both strain backgrounds, and confirmed the deletion using PCR. We found that there were no differences in the cytotoxicity of JH2010Δ*N*Δ*cI*Δ*cro* and JH2016Δ*N*Δ*cI*Δ*cro* across all culture fractions, Figure 6B, a finding that suggests that the regulatory region is necessary for RecA-independent localization of toxin into the supernatant. Based on these results, we hypothesize that *stx*_2a_-phage induction, through the involvement of the regulatory region, differentially influences the RecA-independent in vitro cytotoxicity of strains JH2010 and JH2016.

Expression of *stx*_2a_-phage transcripts in Δ*recA* deletion strains. Attempts to enumerate phage production from bacterial culture supernatants either by plaque assay or qPCR detection of circular phage genomes for JH2010 and JH2016 wild-type and Δ*recA* mutant strains were unsuccessful. Therefore, we measured the level of *stx*_2a_-phage induction in the Δ*recA* mutant strains by conducting qPCR to detect phage transcripts associated with the lytic cycle. We designed primers to detect transcripts for the *stx*_2a_-phage repressor (*cI*, early gene), antiterminator Q (*q*, late-early gene), and Stx2 (*stx*_2_, late gene), Figure 7A. Since the two sequences have low nucleotide identity, two different sets of primers were designed to detect *cI* transcripts from the JH2010 *stx*_2a_-phage (*cI*_JH2010_) and JH2016 *stx*_2a_-phage (*cI*_JH2016_). We adjusted the detected Ct values based on calculated primer efficiencies in order to directly compare the two sets of *cI* primers (see Table 2). *cI*_JH2010_ and *cI*_JH2016_ were only detected in JH2010Δ*recA* and JH2016Δ*recA,* respectively, Figure 7B. We found that *q* transcript levels were significantly higher in JH2010Δ*recA*, Figure 7B, a finding that suggests that there is more *stx*_2a_-phage induction in JH2010Δ*recA* than in JH2016 Δ*recA*, in agreement with our findings of elevated toxin levels in Figure 6. However, both strains had similar levels of *stx*_2_ transcript, a result that may mean the *stx*_2_ transcript is unstable, or that there is an additional level of regulation at the translational level. Alternatively, because the Vero assay is quite sensitive, we may be able to detect differences in Stx levels that are not identified in the qPCR assay. Finally, the *cI*_JH2016_ transcript levels from JH2016Δ*recA* trended higher than *cI*_JH2010_ levels from JH2010Δ*recA,* but the difference was not significant, Figure 7B. The wild-type strains exhibited similar trends in the transcript levels of *cI*_JH2010_/*cI*_JH2016_ and *q* to those observed in the Δ*recA* mutant strains, but the differences were not significant, Appendix A. Our data suggest that JH2010Δ*recA* exhibits transcript levels associated with increased *stx*_2a_-phage induction, compared to JH2016Δ*recA*. These results support the hypothesis that RecA-independent *stx*_2a_-phage induction is responsible for the differential phenotypes exhibited by JH2010Δ*recA* and JH2016Δ*recA*.

Regulatory protein structure prediction. For lambdoid phages, the proteins encoded within the regulatory region act as switches to regulate the phage life cycle. The organization and predicted NCBI annotations of the proteins encoded within the *stx*_2a_-phage regulatory region suggest that the *stx*_2a_-phage regulatory proteins identified in this study function similarly to the lambda phage regulatory proteins, despite the low amino acid homology. Since JH2010 and JH2016 exhibited differential cytotoxicity in studies involving the phage regulatory region and *recA*, we analyzed the amino acid sequence of the phage repressor CI (encoded by *cI*), which is the only *stx*-phage regulatory protein known to interact directly with RecA. Based on pairwise alignment by Clustal Omega, the CI protein sequences from the JH2010 *stx*_2a_-phage (CI_2010_), JH2016 *stx*_2a_-phage (CI_2016_), and lambda phage (CI_λ_) showed little amino acid identity, Figure 8A. Additional protein domain and residue analyses using InterProScan found that all of the CI sequences shared identical functional protein domains, conserved catalytic site residues, and similar localization of DNA-binding residues, despite the divergence in amino acid sequences, Figure 8A. Combined, these data suggest that, despite the low amino acid sequence homology, CI_JH2010_ and CI_JH2016_ exhibit similarities in domain organization to each other (and to CI_λ_) and contain highly conserved residues involved in DNA-binding and autoproteolytic functions.

We next used Phyre2 to predict the secondary and tertiary structure of the three CI proteins. Using the known crystal structure of CI_λ_ as a reference, we generated predicted structures for CI_JH2010_ and CI_JH2016_ with greater than 99% confidence. We also analyzed the CI_λ_ sequence as a test for the Phyre2 algorithm, and found that the results were similar to the published CI_λ_ crystal structure [24]. Visualization of these predicted Phyre2 structures showed similarities in the three-dimensional organization for all three CI proteins, Figure 8B. However, there were slight differences in structure within the functional domains of the proteins, including the relative spatial localization of residues important for the two main functions of CI: DNA-binding and auto-proteolysis, Figure 8B. Taken together, these data indicate key functional similarities and structural differences between the *stx*_2a_-phage CI proteins from JH2010 and JH2016, leading us to hypothesize that CI_JH2010_ and CI_JH2016_ may exhibit the same general function as lambdoid phage repressors, but likely have differential efficacy.

## 4. Discussion

In this study, we found that O157:H7 STEC strains may produce relatively high levels of Stx2 in the absence of *recA*, and that strains with different *stx*_2a_-phage CI types show differences in RecA-independent Stx2 localization and RecA-dependent virulence in mice. Similar to previous studies, deletion of *recA* in JH2010 and JH2016 significantly reduced Stx2 production compared to the wild-type strain; however, toxin levels were still above 10^3^ CD_50_/mL culture [14]. Unexpectedly, JH2010Δ*recA* had higher levels of in vitro supernatant cytotoxicity and in vivo fecal cytotoxicity from infected mice, as compared to JH2016Δ*recA*. Comparisons of the *stx*_2a_-phage regulatory region identified sequence differences in key phage regulatory genes, and our data support a role for this region in differential, RecA-independent Stx2 production in the studied strains. Wild-type strains JH2010 and JH2016 did not exhibit significant differences in Stx2 production in vitro, but did show differences in in vivo Stx2 production. The difference in the amount of in vivo Stx2 production may also implicate the *stx*_2a_-phage regulatory region in differential RecA-mediated phage induction. The finding that STEC *recA* mutant strains are capable of producing enough Stx in vivo to cause modest morbidity in Str-treated mice has implications for potential STEC therapeutics that target RecA-mediated *stx*-phage induction [13,25].

The *stx*_2a_-phage regulatory regions were non-homologous between JH2010 and JH2016. However, the synteny of the genes within the region and the predicted protein domains of the gene products suggested shared a functionality as lambdoid phage regulatory regions. Based on transcript levels and protein structure predictions, we hypothesize that differences in the *stx*_2a_-phage repressor CI may differentially influence RecA-independent *stx*_2a_-phage induction, and consequently, Stx2 production. Previous work with purified CI proteins from various *stx*- and non-*stx*-encoding lambdoid phages show functional differences between the proteins [26,27]. Additional studies with purified *stx*-phage CI proteins may help us understand how differences in the structure and function of various *stx*-phage repressors can influence their regulatory activity in vitro and in vivo.

Our results support a recent study on the phylogeny of *stx*-phage regulatory regions and the encoded proteins. When studying *stx*-encoded CI sequences from the NCBI virus database and various European STEC bio-projects, eight clades of CI sequences were identified [28]. Based on that clade classification scheme, the *stx*_2a_-phage CI protein sequences from JH2010 and JH2016 can be classified as clade V (CI_V_) or VII (CI_VII_), respectively. The same study found that CI_V_ was primarily encoded on Stx1-phages, while CI_VII_ was more commonly encoded on Stx2-phages [28]. In our study, we found that CI_V_ and CI_VII_ were both encoded on *stx*_2a_-phages. When we interrogated the NCBI PulseNet STEC genome reference library, we found that the CI_VII_ from JH2016 was more prevalent than CI_V_ from JH2010, and, similar to the previous study, found that CI_V_ was usually associated with *stx*_1_-phages, Appendix A. The findings from this study indicate that the genetic mosaicism and recombination found among many lambdoid phages [29,30] also applies to *stx*-phages isolated from patients. Finally, our findings suggest that the mosaicism of *stx*-phage regulatory sequences, and possible differences in CI function, influence downstream Stx production and could be used in the future to predict the virulence of STEC clinical isolates.

## Figures and Tables

**Figure 1 microorganisms-11-01925-f001:**
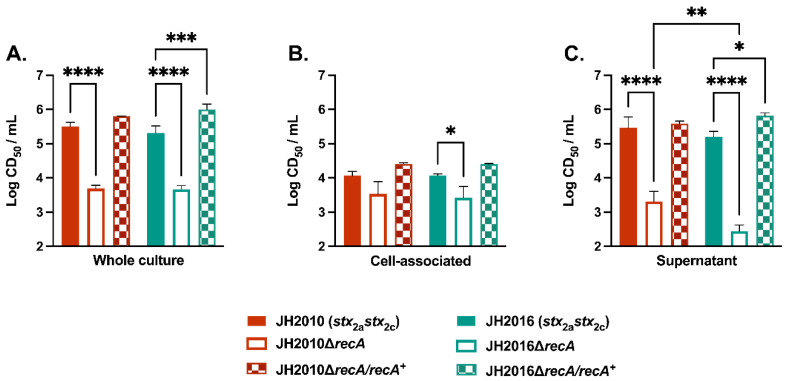
In vitro cytotoxicity of JH2010 and JH2016 wild-type and Δ*recA* mutant strains. Samples were taken from overnight cultures grown in LB-YE from (**A**) the whole culture, or divided into (**B**) cell-associated or (**C**) supernatant fractions. Results are shown as mean log (CD_50_/mL) ± SD (*n* = 3 biological replicates). Statistical analyses were conducted by one-way ANOVA with Šídák’s test for multiple comparisons for each fraction. * *p* < 0.05, ** *p* < 0.01, *** *p* < 0.001, **** *p* < 0.0001.

**Figure 2 microorganisms-11-01925-f002:**
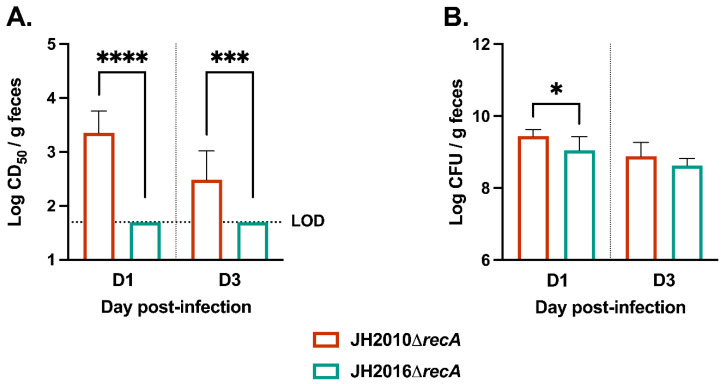
Cytotoxicity and colonization from mice infected with Δ*recA* mutant derivative strains of JH2010 or JH2016. Str-treated mice were infected with either JH2010Δ*recA* or JH2016Δ*recA*. Fecal samples were collected on days 1 and 3 post-infection and assayed for (**A**) fecal cytotoxicity, shown as mean log_10_(CD_50_/g feces) ± SD, and (**B**) colonization, shown as mean log_10_(CFU/g feces) ± SD. (*n* = 5 biological replicates) Statistical analyses were conducted by unpaired *t*-test for each day post-infection. * *p* < 0.05, *** *p* < 0.001, **** *p* < 0.0001. LOD = limit of detection.

**Figure 3 microorganisms-11-01925-f003:**
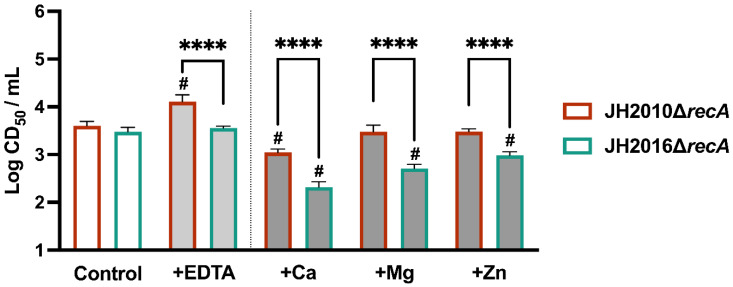
Influence of RecA-independent inducing agents on the cytotoxicity from whole culture samples from JH2010Δ*recA* or JH2016Δ*recA*. Overnight cultures of JH2010Δ*recA* and JH2016Δ*recA* were grown in LB-YE media (as control) or LB-YE supplemented with 50 μM EDTA, 20 mM CaCl_2_, 20 mM MgCl_2_, or 0.5 mM ZnCl_2_. Significance was calculated by one-way ANOVA with Šídák’s test for multiple comparisons (*n* = 3 biological replicates). Comparisons between JH2010Δ*recA* and JH2016Δ*recA* for each culture condition: **** *p* < 0.0001. Comparisons between the cytotoxicity of control cultures and cultures with additions for each strain: # *p* < 0.0001.

**Figure 4 microorganisms-11-01925-f004:**
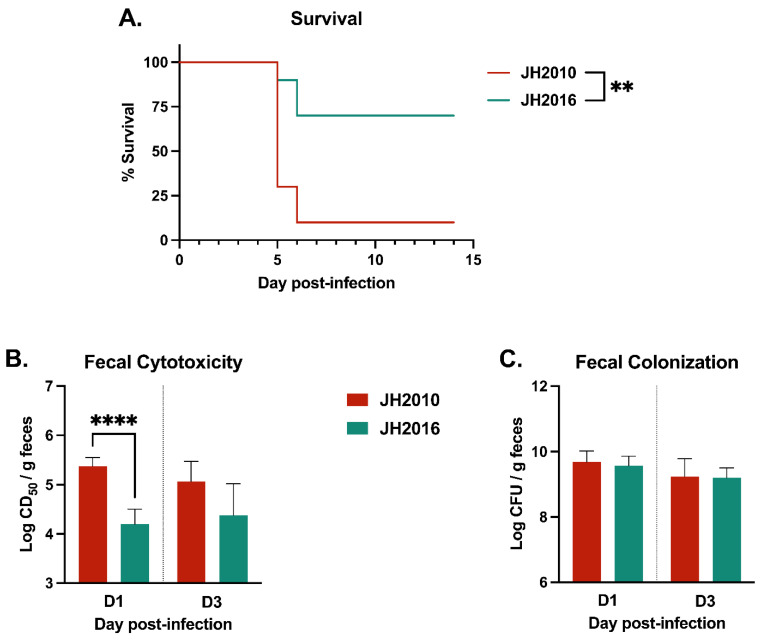
Mouse infection with wild-type JH2010 or JH2016 at an intermediate dose. (**A**) Survival of Str-treated mice infected with 10^6−7^ CFU/mouse of JH2010 or JH2016 (*n* = 10 mice per strain). Fecal samples collected on days 1 and 3 post-infection (*n* = 5 mice per strain) were used to determine (**B**) fecal cytotoxicity, shown as mean log (CD_50_/g feces) ± SD, and (**C**) fecal colonization shown as mean log (CFU/g feces) ± SD. Statistical significance was calculated by (**A**) log-rank test or by (**B**,**C**) unpaired *t*-test on each day post-infection. ** *p* < 0.05, **** *p* < 0.0001.

**Figure 5 microorganisms-11-01925-f005:**
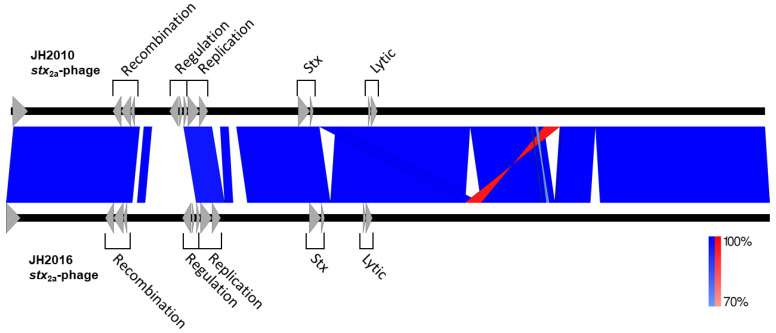
Genomic comparisons of the assembled *stx*_2a_-phages from JH2010 and JH2016 using NCBI Nucleotide BLAST. Alignment of the JH2010 and JH2016 *stx*_2a_-phage sequences was visualized using Easyfig. Gray arrows represent relevant functional clusters of genes based on gene product annotation within the labelled phage regions. The saturation of the color indicates percent homology between the two sequences; blue blocks represent sequences in the same orientation, while red represents sequences in the opposite orientation. White regions represent areas with less than 70% identity. The JH2010 and JH2016 *stx*_2a_ phage genome sizes are 61,291 or 63,000 nucleotides, respectively.

**Figure 6 microorganisms-11-01925-f006:**
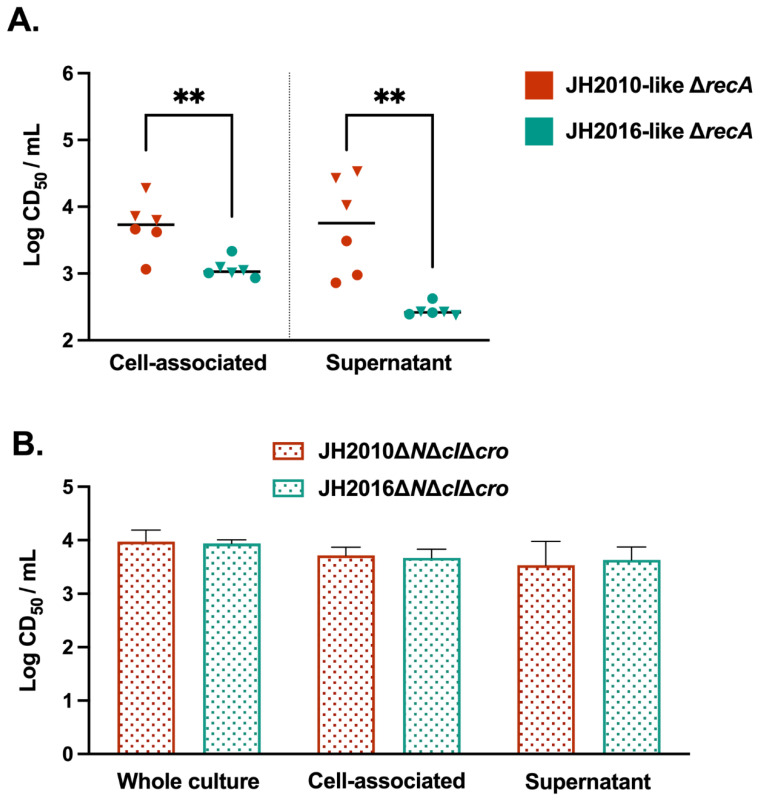
Influence of the *stx*_2a_-phage regulatory region on Stx2 production in *stx*_2a_*stx*_2c_ O157:H7 strains. (**A**) Overnight cultures of *recA* mutant strains were divided into cell-associated or supernatant fractions. Parental strains with JH2010-like or JH2016-like *stx*_2a_-phage regulatory sequences were identified from either PulseNet (triangles) or other NCBI databases (circles). (**B**) Samples were taken from the whole culture or divided into cell-associated or supernatant fractions for strains JH2010Δ*N*Δ*cI*Δ*cro* and JH2016Δ*N*Δ*cI*Δ*cro*. Results are shown as mean log (CD_50_/mL) ± SD, (*n* = 3 biological replicates). Statistical analyses were conducted by (**A**) unpaired *t*-test for each fraction or (**B**) one-way ANOVA with Šídák’s test for multiple comparisons for each fraction. ** *p* < 0.01.

**Figure 7 microorganisms-11-01925-f007:**
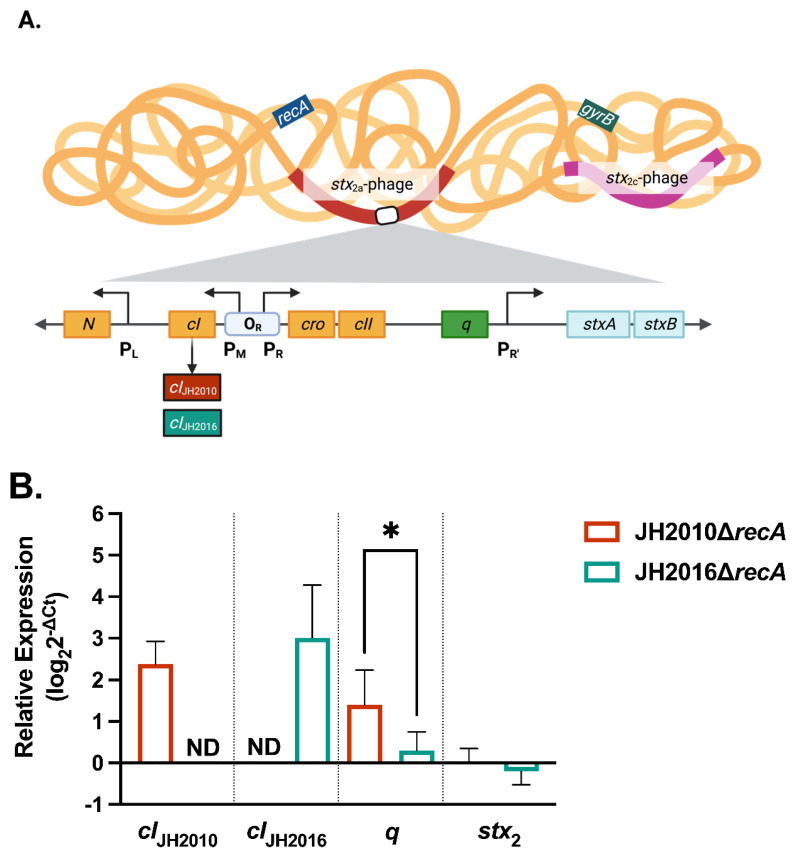
Gene expression of Δ*recA* strains by RT-qPCR. (**A**) Location and order of genes on a STEC chromosome or *stx*_2a_-phage, not to scale. (**B**) Relative transcription of genes from overnight LB-YE cultures of JH2010Δ*recA* and JH2016Δ*recA*, as measured by RT-qPCR. Results are graphed as mean log fold change from *gyrB* transcript ± SD. Significance was calculated by one-way ANOVA with Šídák’s test for multiple comparisons (*n* = 7 biological replicates). * *p* < 0.05.

**Figure 8 microorganisms-11-01925-f008:**
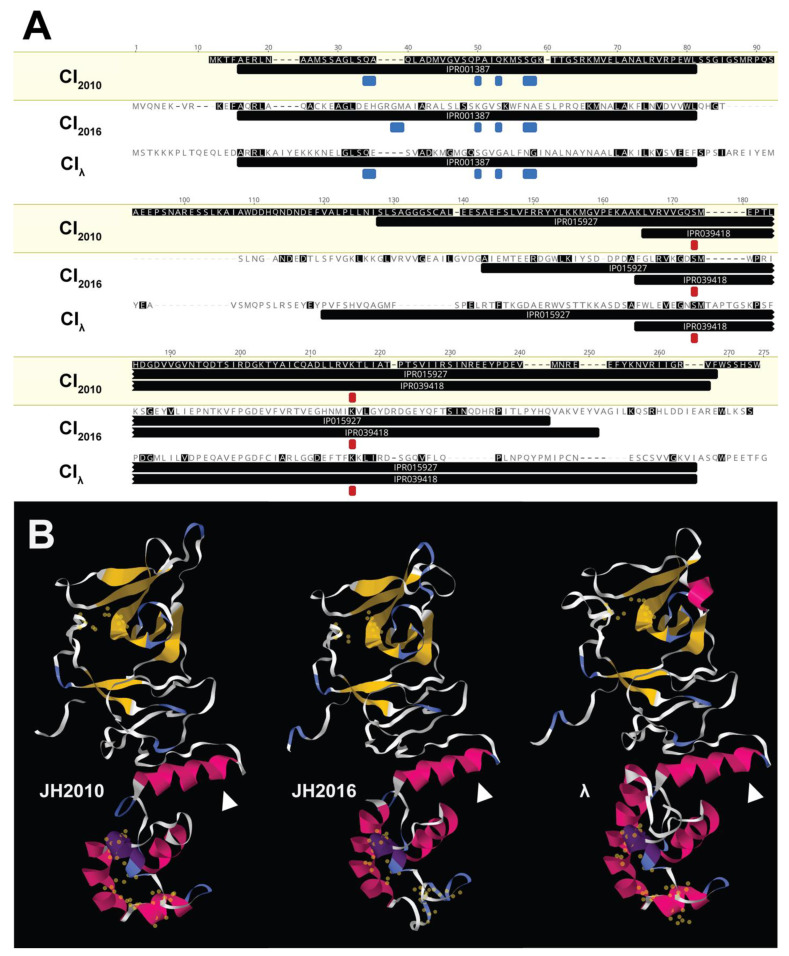
The CI_JH2010_, and CI_JH2016_, and CI_λ_ amino acid sequences and predicted protein structures. (**A**) Multiple sequence alignment of CI sequences by Clustal Omega from the JH2010 *stx*_2a_-phage, JH2016 *stx*_2a_-phage, and lambda phage. Amino acids highlighted in black show agreement with the reference CI_JH2010_ sequence. The black annotations below each sequence were determined by InterProScan, and represent the protein domains identified from the InterPro database. DNA-binding residues (blue) and the serine protease catalytic site residues (red) are also marked. (**B**) Tertiary protein structure predictions of CI_JH2010_ and CI_JH2016_ as determined by Phyre2 using CI_λ_ as a reference (PDB #3BDN). The ribbon diagram is color-coded, based on secondary structure. All structures are shown with the C terminal region in the upper half of the image, and were oriented for viewing based on the noted N terminal helix (white arrow). Identified catalytic and DNA-binding residues were highlighted in the C and N terminal regions, respectively (yellow spheres).

**Table 1 microorganisms-11-01925-t001:** Bacterial strains used in this study.

Strain Name	Description	NCBI Accession # (Parental Str^s^ Strain, Unless Noted)	Reference
JH2010	Str^r^ isolate of CDC #06-3462	NZ_CP034794.1, OP797663 *	[14]
JH2016	Str^r^ isolate of CDC #2009c-4687	NZ_CP034799.1	[14]
JH2010Δ*recA*	*recA* replaced with *cat* cassette in JH2010	N/A	[14]
JH2016Δ*recA*	*recA* replaced with *cat* cassette in JH2016	N/A	This study
pJH206	pACYC177 vector with *recA* + 500 bp upstream	N/A	[14]
JH2010Δ*recA*/*recA*^+^	Complementation of JH2010Δ*recA* with pJH206	N/A	[14]
JH2016Δ*recA*/*recA*^+^	Complementation of JH2016Δ*recA* with pJH206	N/A	This study
JH2010Δ*N*Δ*cI*Δ*cro*	*stx*_2a_-phage region from *N* to *cro* replaced with *cat* cassette in JH2010	N/A	This study
JH2016Δ*N*Δ*cI*Δ*cro*	*stx*_2a_-phage region from *N* to *cro* replaced with *cat* cassette in JH2016	N/A	This study
JH2011Δ*recA*	*recA* replaced with *cat* cassette in JH2011 (Str^r^ isolate of CDC #08-3914)	NZ_CP034808.1	This study
JH2017Δ*recA*	*recA* replaced with *cat* cassette in JH2017 (Str^r^ isolate of CDC #2010c-3142)	NZ_CP034801.1	This study
PA28-Str^r^Δ*recA*	*recA* replaced with *cat* cassette in RA10 (Str^r^ isolate of strain PA28)	NC_041935.1 *	This study, parental strain [15]
TW14359-Str^r^Δ*recA*	*recA* replaced with *cat* cassette in RA13 (Str^r^ isolate of strain TW14359)	NC_013008.1	This study, parental strain [16]

* *stx*_2a_-phage sequence only.

**Table 2 microorganisms-11-01925-t002:** Primers used in this study.

Primer/Primer Pairs	Sequence	Reference
Primers for mutagenesis and confirmation of deletion strains		
recA-KO_F	CTCATTTTATGGCTATCGACGAAAACAAACAGAAAGCGTTGGCGGCAGCACTGGGCCAGCACGTAAGAGGTTCCAACTTTCACCATAATG	[14]
recA-KO_R	ACGAACGATTAAAAATCTTCGTTAGTTTCTGCTACGCCTTCGCTGTCATCTACAGAGAAATTACGCCCCGCCCTGCCACTCATC	[14]
2010_N-CI-Cro-KO_F	TTAACATAAATAGTCTTTTTCACCATAAGCATACTCAATAAGTCCATACGGCACGTAAGAGGTTCCAACTTTCACCATAATG	This study
2010_N-CI-Cro-KO_R	TTATTCAGCCTGGTTCGGATGAGGAAACAGGTGCGGTAAGTCCGGGCGAATTACGCCCCGCCCTGCCACTCATC	This study
2016_N-CI-Cro-KO_F	TCACCTCGCCGTCAGTTGTTTTGATTTCCGGTAGCCTGCCGCGTAAATGGGCACGTAAGAGGTTCCAACTTTCACCATAATG	This study
2016_N-CI-Cro-KO_R	TTATGCAGCCAGAAGGTTCTTTTTGCTTATTTCAAGCATTTCGCTTGCTTTTACGCCCCGCCCTGCCACTCATC	This study
ScreenRecA_upF	TAACAACTGCCGAGTCTTGTACCGG	This study
ScreenN_upF	CTGTCGCCTGGAATCTCC	This study
ScreenCm_R	CACGACGATTTCCGGCAGTTTC	[14]
qPCR primers qPCR was conducted with the following thermocycler conditions: 95 °C for 15 min, followed by 40 cycles of: 94 °C for 15 s, 52 °C for 30 s, and 72 °C for 30 s.		
gyrB_rtF/gyrB_rtR	F: GGTAGATAACGCTATCGACG R: GGGTGAATACCGGTCGG	This study
CI-2010_rtF/CI-2010_rtR *	F: GAGTCCGCAGAATTCTCTC R: CAACCCTGACCAACTTTGC	This study
CI-2016_rtF/CI-2016_rtR *	F: GCGCTTGCGAAATTTCTAAAC R: CCATCAACACCAAGAATTGC	This study
Q3_rtF/Q3_rtR	F: GACTGATCCCCGAAAAAGTA R: CAACCAGCAAGTCATGCAG	[21]
stx2_rtF/stx2_rtR	F: CGGCGGATTGTGCTAAAG R: GGTACTGGATTTGATTGTGACAG	This study

* CI-2010 primer efficiency~100%, CI-2016 primer efficiency~94%.

## Data Availability

The data presented in this study are available in the article and Appendix A.

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
