# Peer review of "Differences in the Shiga Toxin (Stx) 2a Phage Regulatory Switch Region Influence Stx2 Localization and Virulence of Stx-Producing Escherichia coli in Mice"

_microorganisms, 2023, doi:10.3390/microorganisms11081925_

Round 1

Reviewer 1 Report

Shiga toxins encoded on Lambdoid-like prophages in E. coli are a major cause of serious food-borne illnesses globally. It has been shown that some STEC strains produce Stx even in the absence of RecA (which triggers the lytic switch of Lambda prophage) making the mechanisms regulating Stx production important to understand. Kudos to Atitkar and Melton-Celsa for this well-written and presented manuscript.

My concerns/questions:

-        There is a lack of details in the Methodology that would be required to make the work reproducible, see below. If there is a word limit for the Methods potentially the authors could consider including further details in supplementary material.

-      Mice were given streptomycin prior to the STEC strain lavage, presumably to remove all non-STEC bacteria that are sensitive to Str, does that totally preclude the possibility that there may be other organisms in the mice microbiomes that are Strr that might contribute to the CFU/g counts?

-        Consider rephrasing the title, phage genomes have many regulatory regions/elements and it is somewhat ambiguous. Not all readers of Microorganisms will be familiar with Lambda-like/Stx-prophages.

Methods

Section 2.1

Line 84 -An exact length of time for overnight culture growth should be included. Volumes should also be included.

Did each culture derive from a single colony?

How many replicates of all the overnight cultures were made (possibly 3 as Figure 1 and another has n=3)? Were the replicates biological or technical replicates?

Were cultures stored on ice/refrigerated prior to any assay/centrifugation step, if so that should be included and roughly what was the time for each?

Line 87 – provide aliquot volumes

Include centrifugation details (time, rpm, temp) for cell pelleting and clearing of homogenized cells

Section 2.2

Lines 107-108 – include details for any centrifugation step between homogenization and plating of the supernatant

Line 109, what concentrations of STR and CAM were employed?

Section 2.3

Line 117-118 – were the PCR products used to confirm the RecA and the prophage regulatory region deletions sequenced?

Results

Section 3.1 – were the values for cytotoxity adjusted for the differing amounts of CFU/ml per sample (even though it is noted that there were only modest differences in growth), dry weight or in any way? It is intriguing that there are the variations in OD600 that was not observed in the cells counts.

Do the cytotoxity values for the cell pellet and supernatant sum to a value that is close to the cytotoxicity value of the whole culture for each strain?

Section 3.6 – no phage particles were enumerated, was this due to the absence of a suitable indicator strain (if so, were the strains with CI deleted tested?). Was RT-qPCR conducted on any genes whose products are incorporated into the virion?

Figure 3. – are the cytotoxicity levels described for whole cell samples after homogenization?

Figures 5 and S4

– The genome comparison figures should contain some sort of indicator of genome length and/ or ruler

-        Inclusion of promoters would be helpful, the reader is left presuming the Stx share the same promoter as the virion genes.

-        The regions shaded red are confusing (the description of those regions is as blocks in the legend).

-        The shading for sequence identity is not clear with the cut-off at 70% similarity so that the matches for homologous region look like they are either 100 % similar or not at all (although they are mostly >90 % or <70 % similar).

-        Is “Lytic” meant to indicate Lysis genes or generally genes associated with the lytic life-cycle??

-        Similarly, what does “lytic” mean in the text, e.g., lines 264 and 265

-        What is the nucleotide similarity of each of the lysis genes between JH2010 and JH2016?

Lines 285-289 – there appears to be higher variability between the cytotoxicity levels of the replicates for the strains with JH2010-like Stx prophages, versus those with JH2016-like prophages. Any possible suggestions as to why that might be?

Supplementary Fig 4. Are there any other genes in the region encompassing N-CI-Cro that were deleted?

Author Response

We thank the reviewer for their time, and have attached our response. 

Reviewer 2 Report

In this study, the authors report that distinct strains of shiga toxin-producing E. coli (STEC) show differential RecA-independent production of Shiga toxin subtype 2 (Stx2), both in vitro (LB-YE) and in vivo (mice). The phenotype of higher production of Stx2 correlated with a higher mortality induced in mice. The authors found that the differential expression of Stx2 depends on the presence of the stx2a-phage region encoding regulatory factors including N, CI, and Cro proteins, but the specific effect leading to this differential Stx2 expression remains to be determined.

The manuscript is well written and major conclusions are in general supported by data.

Comments.

-Major conclusion of this study will be reinforced by data from complementation assays of the mutants lacking the stx2a-phage regulatory region (Fig. 6B). Even with the complementation with the non-homologous regulatory region (the region from the other strain.

-Discuss about how the stx2a-phage regulatory region could mediate the differential Stx2 expression between the strains tested. It seems that this regulatory region is repressing expression of Stx2 in the JH2016 strain (Fig. 6B). Is it right?

-Lines 182-184. It is not clear for me this conclusion based on results from figure S2. I do not see the referred transient decrease in the weight of infected mice.

-Lines 329-330. This conclusion is not convincing. Results from figure 7B show a difference only for the q transcripts levels between the strains in study. How this difference in the q transcripts would lead to a difference in Stx2 expression? Link the results from figure 7B with those from figure 6B and with the putative function of CI and q factors.

Author Response

We thank the reviewer for their time in reviewing the manuscript. Our responses are attached. 

Reviewer 3 Report

The manuscript of Atitkar and Melton-Celsa studies the RecA-induction of Stx in STEC. It is a well written manuscript that contains a lot of information, and it is well thought out. Despite I consider that it could be published as it is, here are some comments and suggestions for the authors that may (or not) improve it:

In section 3.3 the authors show the effect of adding EDTA or different divalent cation salts in the cytotoxicity of the recA strains. However, what would happen in the WT strain?

Also, in figure 3, the authors compare the effects of the different compounds between strains. But it is not clear that the authors also have determined the statistics between treatments. Yes, it is written in the figure legend, but it is not obvious. I wonder if it would be more helpful to plot it differently.

In section 3.6 the authors determine the expression levels of cI, q and stx only at the ΔrecA strain. Considering that the authors claim that it is a RecA-independent effect, adding the qPCR with the WT strain (and seeing the same as in the ΔrecA strain) may give more strength to their findings.

In section 3.8 the authors show the predicted 3D structure of the different CI. Even though they are quite similar the authors point out several differences in the functional domains (line 363-365). I would recommend improving the labelling in the figure (the yellow dots are difficult to see) and labelling the different domains.

Author Response

We thank the reviewer for their time in considering this manuscript. Our responses are attached. 
